# CRISPR-Knockout of *CSE* Gene Improves Saccharification Efficiency by Reducing Lignin Content in Hybrid Poplar

**DOI:** 10.3390/ijms22189750

**Published:** 2021-09-09

**Authors:** Hyun-A Jang, Eun-Kyung Bae, Min-Ha Kim, Su-Jin Park, Na-Young Choi, Seung-Won Pyo, Chanhui Lee, Ho-Young Jeong, Hyoshin Lee, Young-Im Choi, Jae-Heung Ko

**Affiliations:** 1Department of Forest Bio-Resources, National Institute of Forest Science, Suwon 16631, Korea; hajang@korea.kr (H.-A.J.); baeek@korea.kr (E.-K.B.); sujin2419@korea.kr (S.-J.P.); hyoshinlee@korea.kr (H.L.); 2Department of Plant & Environmental New Resources, Kyung Hee University, Yongin 17104, Korea; minha123@khu.ac.kr (M.-H.K.); nayoung430@khu.ac.kr (N.-Y.C.); swpyo18@khu.ac.kr (S.-W.P.); chan521@khu.ac.kr (C.L.); ratank@khu.ac.kr (H.-Y.J.)

**Keywords:** biofuels, caffeoyl shikimate esterase (CSE), CRISPR/Cas9, hybrid poplar, lignin, saccharification

## Abstract

Caffeoyl shikimate esterase (CSE) has been shown to play an important role in lignin biosynthesis in plants and is, therefore, a promising target for generating improved lignocellulosic biomass crops for sustainable biofuel production. *Populus* spp. has two *CSE* genes (*CSE1* and *CSE2*) and, thus, the hybrid poplar (*Populus alba* × *P. glandulosa*) investigated in this study has four *CSE* genes. Here, we present transgenic hybrid poplars with knockouts of each *CSE* gene achieved by CRISPR/Cas9. To knockout the *CSE* genes of the hybrid poplar, we designed three single guide RNAs (sg1–sg3), and produced three different transgenic poplars with either *CSE1* (CSE1-sg2), *CSE2* (CSE2-sg3), or both genes (CSE1/2-sg1) mutated. CSE1-sg2 and CSE2-sg3 poplars showed up to 29.1% reduction in lignin deposition with irregularly shaped xylem vessels. However, CSE1-sg2 and CSE2-sg3 poplars were morphologically indistinguishable from WT and showed no significant differences in growth in a long-term living modified organism (LMO) field-test covering four seasons. Gene expression analysis revealed that many lignin biosynthetic genes were downregulated in CSE1-sg2 and CSE2-sg3 poplars. Indeed, the CSE1-sg2 and CSE2-sg3 poplars had up to 25% higher saccharification efficiency than the WT control. Our results demonstrate that precise editing of *CSE* by CRISPR/Cas9 technology can improve lignocellulosic biomass without a growth penalty.

## 1. Introduction

Plant lignocellulosic biomass (i.e., wood) is an important renewable and sustainable feedstock for the production of both biomaterials and biofuels [1,2]. The production of biofuels from biomass is gaining more attention due to the growing global climate crisis [3,4].

Polysaccharides in biomass are fermented into ethanol or other compounds by optimized microorganisms after saccharification [5]. However, biomass does not easily decompose due to the complex chemical and physical structure of the plant cell wall, which is referred to as biomass recalcitrance [6,7,8,9]. One of the major causes of biomass recalcitrance is the presence of lignin, a phenolic polymer that provides strength and hydrophobicity to the secondary cell wall. Lignin impedes the efficient enzymatic degradation of cellulose and hemicellulose into fermentable sugars by immobilizing hydrolytic enzymes and physically restricting access to the polysaccharide substrate [6,8,10,11].

Lignin is a heterogeneous polymer comprising three types of monomers synthesized in the phenylpropanoid pathway starting with the aromatic amino acid, phenylalanine. After deamination of phenylalanine by phenylalanine ammonia-lyase (PAL), the resulting cinnamic acid undergoes a series of aromatic ring and propene tail modifications resulting in three hydroxycinnamoyl alcohols with different degrees of methoxylation, namely *p*-coumaryl, coniferyl, and sinapyl alcohols. Once incorporated into the polymer, these monolignols produce *p*-hydroxyphenyl (H), guaiacyl (G), and syringyl (S) units, respectively [12,13].

A number of pretreatment methods have been developed to lower biomass recalcitrance, but pretreatment is still a relatively expensive step in the manufacturing process of biofuels [14,15]. Thus, bioengineering of trees that produce less lignin but maintain normal growth would reduce processing costs and the carbon footprint of biofuel production [7,16,17,18,19].

Recently, Vanholme et al. [15] demonstrated that caffeoyl shikimate esterase (CSE) catalyzes the conversion of caffeoyl shikimate into caffeate in *Arabidopsis*, which bypasses the second hydroxycinnamoyl-CoA shikimate/quinate hydroxycinnamoyltransferase (HCT) reaction with 4-coumarate:CoA ligase (4CL) in a lignin biosynthetic pathway [20]. Loss of function of *CSE* by T-DNA insertion in *Arabidopsis* resulted in a reduction of lignin levels by up to 36% with preferential accumulation of H units (30-fold) [15]. A similar phenotype was reported in a *CSE* loss-of-function mutant of *Medicago truncatula* generated by transposon insertion [21]. Saleme et al. [22] later demonstrated that downregulation of *CSE* by RNAi silencing resulted in a reduction in lignin deposition (up to 25%) with increased levels of H units (two-fold) in the lignin polymer and a higher cellulose content in hybrid poplar (*Populus tremula* × *P. alba*). Recently, *LkCSE* was successfully cloned from the gymnosperm tree species, *Larix kaempferi*, and was shown to be able to convert caffeoyl shikimate to caffeate and shikimate by in vitro assays using recombinant LkCSE protein [23].

In both *Arabidopsis* and hybrid poplar, saccharification efficiency can be dramatically increased by mutation of *CSE* due to the reduction of lignin deposition. However, the overall plant growth was not severely inhibited [15,22]. These results suggest that *CSE* is not only important for lignin biosynthesis but is also a promising target for generating improved lignocellulosic biomass crops for biofuel production [15,22].

In this study, we functionally characterized transgenic *CSE*-knockout hybrid poplars generated by clustered regularly interspaced short palindromic repeat (CRISPR)/CRISPR-associated protein 9 (Cas9) technology. CRISPR/Cas9 technology is based on the Cas9 nuclease and single-guide RNA (sgRNA) for target DNA sequence recognition, and can be utilized to make gene-specific insertion or deletion (indel) mutations [24]. CRISPR/Cas9 has been widely used for genome editing in plants due to its great efficiency and simplicity [25,26,27,28]. We designed sgRNAs for *CSEs* in the hybrid poplar (*Populus alba* × *P. glandulosa*, clone BH) and produced transgenic CSE-CRISPR poplar knockouts of either *CSE1* (i.e., CSE1-sg2) or *CSE2* (i.e., CSE2-sg3), or both genes; mutation of either CSE1 or CSE2 resulted in a reduction in lignin deposition by up to 29.1% and significantly increased saccharification efficiency (up to 25%). We will discuss the significance of using this approach to improve woody biomass feedstock for biofuel production.

## 2. Results

### 2.1. Production of Transgenic Hybrid Poplars with CRISPR-Knockout of CSE Genes

Two homologous *CSE* genes are present in the genome of *Populus trichocarpa*, namely *PtrCSE1* (Potri.001G175000) and *PtrCSE2* (Potri.003G059200). These genes have 91% amino acid sequence identity to each other, and around 80% to *Arabidopsis CSE* (At1g52760; Appendix A). The hybrid poplar (*Populus alba* × *P. glandulosa*, clone BH) used in this study has four *CSE* genes. Two *CSE* genes (*PaCSE1* and *PaCSE2*) come from the *P. alba* genome, while the other two (*PgCSE1* and *PgCSE2*) are from the *P. glandulosa* genome (Appendix A). To precisely knockout each *CSE* gene in the hybrid poplar, we used CRISPR/Cas9 genome editing technology (see Methods). First, we designed three single guide RNAs (sg1, sg2, and sg3); sg1 was designed to knockout all four *CSE* genes by targeting 1st exon; sg2 and sg3 were designed to knockout *CSE1* and *CSE2*, respectively, by targeting the 2nd exon (Figure 1a; Appendix A). Then, we produced transgenic CSE-CRISPR hybrid poplars (CSE1/2-sg1, CSE1-sg2, and CSE2-sg3) using the vector constructs of each sgRNA.

Next, we examined the resulting mutations at the *CSE* loci of the regenerated transgenic CSE-CRISPR hybrid poplars by PCR amplification and targeted deep sequencing, (Illumina MiniSeq; see Methods) and summarized (Figure 1b). Mutation frequencies in the lines of CSE1/2-sg1, CSE1-sg2 and CSE2-sg3 poplars were 91.7%, 80.0%, and 96.3%, respectively (Figure 1b). CSE1/2-sg1 poplars showed the highest biallelic/homozygous mutations (95.5%). On the other hand, the biallelic/homozygous mutation of CSE1-sg2 poplars was the lowest (20.8%) but the monoallelic and chimeric mutations were relatively higher (45.8% and 33.4%, respectively) than those of CSE1/2-sg1 and CSE2-sg3 poplars. Among these mutants, we selected a total of seven representative lines of higher indel biallelic/homozygous mutations for further functional characterization (Figure 1c).

### 2.2. Predicted CSE Protein and CSE Gene Expression in Transgenic CSE-CRISPR Hybrid Poplars

To visualize the functional significance of the CRISPR/Cas9-induced mutations in each line, we prepared a schematic diagram of the predicted CSE proteins by querying the gene edited sequences of each transgenic line using the ORF finder program of NCBI (https://www.ncbi.nlm.nih.gov/orffinder/20210817) (Figure 2a). In line #2 of the CSE1/2-sg1 poplar, three *CSE* genes (i.e., *PaCSE1*, *PaCSE2* and *PgCSE2*) were edited in the 1st exon as per our experimental design; thus, N-terminal deleted proteins (PaCSE1 and PaCSE2; 309 amino acids) or one amino acid-deleted protein were predicted (PgCSE2). However, PgCSE1 remained intact (326 amino acids) with no gene editing (Figure 2a). Indeed, CSE1-sg2 poplars (lines 1, 16 and 28) that were targeted for knockout of *CSE1* had nonsense mutations in both *CSE1* genes (*PaCSE1* and *PgCSE1*) in the 2nd exon, resulting in predicted C-terminal truncated CSE1 (PaCSE1 and PgCSE1) proteins with only 146 and 154 amino acids, respectively, while the other two CSE2 proteins (PaCSE2 and PgCSE2) were intact (Figure 2a). CSE2-sg3 poplars (lines 4, 17, and 19) that were targeted for *CSE2* knockout had mutations only of the two *CSE2* genes but not the other two *CSE1* genes, as expected (Figure 2a). Among these lines, line 19 had nonsense mutations in the second exon of the two *CSE2* genes (*PaCSE2* and *PgCSE2*), which would result in C-terminal truncated CSE2 (PaCSE2 and PgCSE2) proteins with only 143 and 174 amino acids, respectively.

To quantify the expressions of *CSE* genes in the CSE1-sg2 and CSE2-sg3 poplars, quantitative real-time PCR (RT-qPCR) was performed using primers amplifying the C-terminal region after target sites of sg2 and sg3 (Figure 1a). As an internal quantitative control, the *PtrACTIN7* (Potri. 001G309500) gene was used. Expression of *CSE1* and *CSE2* in CSE1-sg2 and CSE2-sg3 poplars, respectively, was considerably reduced compared to the expression of these genes in BH poplar (control) (Figure 2b,c). However, as expected, there is no significant change in *PagCSE1* expression in CSE2-sg3 poplar and vice versa (Figure 2b,c). This result is consistent to our previous report of PDS-CRISPR poplar study [29], and can be explained by nonsense-mediated mRNA decay, a surveillance pathway present in all eukaryotes, which eliminates mRNA transcripts containing premature stop codons, reducing gene expression errors [30].

### 2.3. CSE-CRISPR Hybrid Poplars Have Reduced Lignin Deposition

We measured the Klason lignin contents of CSE-CRISPR poplars together with that of the control BH poplar (three-month-old grown in pot), using cell wall materials obtained from stem tissues. As shown in Figure 3a, both CSE1-sg2 and CSE2-sg3 poplars had Klason lignin deposition that was reduced by up to 16 wt% compared to BH. All three lines of CSE1-sg2 poplars (lines 1, 16, and 28) had a similar reduction in lignin content. However, among CSE2-sg3 poplars, only line 19 showed a clear reduction in lignin. Interestingly, CSE1/2-sg1 poplars had no changes in lignin content compared to BH (Figure 3a). We attributed these results to the gene editing results in each CSE-CRISPR poplar line, as shown in Figure 1c and Figure 2a. We focused on line 16 of CSE1-sg2 and line 19 of CSE2-sg3 poplar for further in-depth analyses.

To quantify the compositional changes of the cell wall components, we performed cell wall analysis using line 16 of CSE1-sg2 and line 19 of CSE2-sg3 poplar grown in LMO field for 8 month (Figure 3b). Our results showed the significant reduction of total lignin contents in CSE-CRISPR poplars up to 29.1% compared to BH poplars. This reduction in lignin content is higher than the results in Figure 3a, which may result from different growth conditions (e.g., three months in pots vs. eight months in LMO fields). Interestingly, both cellulose and hemicellulose contents were slightly increased in the CSE-CRISPR poplars, consistently to the previous report [22]. However, there were no significant changes in the contents of the extractives.

### 2.4. CSE-CRISPR Hybrid Poplars Have Collapsed Xylem Vessels with Decreased S-Lignin Content

Because both CSE1-sg2 and CSE2-sg3 poplars showed a significant reduction in lignin content, we examined secondary xylem formation by stem cross-sections. Both CSE1-sg2 and CSE2-sg3 poplars (line 16 and line 19, respectively) exhibited collapses of xylem vessel cells (e.g., irregularly shaped xylem) (Figure 4), which is commonly found in plants that have defective accumulation of secondary wall components (such as cellulose, lignin and xylan) (for a review, [31]). On the contrary, BH poplars showed normal xylem vessel development (Figure 4). This result is consistent with the reduced lignin content in CSE1-sg2 and CSE2-sg3 poplars shown in Figure 3.

Indeed, Wiesner (also known as phloroglucinol-HCl) and Mäule staining of both CSE1-sg2 and CSE2-sg3 poplars revealed weaker red coloration than observed in BH poplars, suggesting a decrease in lignin deposition and S-lignin content, respectively (Figure 4b,c).

### 2.5. Coordinated Expression Changes of Genes Involved in Lignin Biosynthesis

Next, we examined the expression of genes involved in the lignin biosynthetic pathway (Figure 5). As expected, genes upstream of *CSE* showed relatively stable expression levels compared to downstream genes, except *PtrC4H1* and *PtrC4H2* genes (Figure 5a,b). For example, expression of the downstream genes *PtrCCoAOMT1*, and *PtrCCR2* was significantly suppressed in CSE-CRISPR poplars compared to BH control poplars (Figure 5b).

Both *PtrMYB152* and *PtrMYB92* have been shown to regulate secondary cell wall thickening and increase total lignin content in poplars [32,33]. Interestingly, expression of both transcription factor genes was significantly repressed in our CSE-CRISPR poplars (Figure 5c), which may also have contributed to the reduction in total lignin content of the CSE-CRISPR poplars.

### 2.6. Enhanced Saccharification Efficiency of CSE-CRISPR Transgenic Poplars with Normal Growth Performance

Saccharification efficiency of wood materials from CSE-CRISPR poplars was measured by quantifying the amount of glucose released at different incubation times after hot water or alkali (1% NaOH) pretreatment (Figure 6a). We found a significant increase (>25% at 72 h) in glucose release from NaOH-treated CSE-CRISPR poplars (CSE1-sg2 #16) compared to BH poplars (Figure 6a). These results suggest that biomass recalcitrance was reduced and thus glucose release was improved in CSE-CRISPR poplars, most likely due to decreased lignin content and increased fermentable sugars, as shown in Figure 3.

Previously, Vanholme et al. [15] reported that an *Arabidopsis CSE* loss-of-function mutant (*cse*-*2*) exhibited a 40% reduction in plant growth. Furthermore, loss of function of *CSE* in transposon insertion lines of *M. truncatula* resulted in severe dwarfing and altered development [21]. We therefore investigated the overall growth phenotypes (e.g., stem height and diameter growth) of both CSE1-sg2 and CSE2-sg3 poplars compared to BH poplars. Interestingly, we detected no significant differences in growth among CSE1-sg2 and CSE2-sg3 poplars and BH poplars in a living modified organism (LMO) field test conducted over a year covering all four seasons (Figure 6b).

## 3. Discussion

Lignin is essential for the growth and development of terrestrial plants as it contributes to the creation of a very strong secondary cell wall. At the same time, lignin makes it difficult to process plant biomass into fermentable sugars [6,34]. Not only does *CSE* play an essential role in plant lignin biosynthesis, it is also an excellent target for producing improved biomass crops for sustainable biofuel production [15,22]. Here, we described the generation and functional characterization of transgenic hybrid poplars with knockouts of each *CSE* gene by CRISPR/Cas9 technology.

### 3.1. CSE-Knockout Reduces Lignin Deposition in Poplar Stems

We generated three different transgenic hybrid poplars with mutations of either *CSE1* (CSE1-sg2), *CSE2* (CSE2-sg3), or both genes (CSE1/2-sg1). However, we did not observe any phenotypic changes in CSE1/2-sg1 poplars (both *CSE1* and *CSE2* mutated), most likely due to targeting of the N-terminus of the CSE1/2 protein. In fact, CSE1/2-sg1 poplars are expected to have an intact PgCSE1 protein, PgCSE2, with a single amino-acid deletion, and PgCSE1 and PaCSE2 proteins with the 17 N-terminal amino acids deleted, which could all potentially function properly (Figure 2a). In fact, we performed in-depth analyses on five additional lines, that are two biallelic (#11, #12) and three homo lines (#30, #32, #34). However, all those lines showed similar growth performances with no significant changes of lignin deposition compared to BH poplars (data not shown). Thus, we focused on characterizing CSE1-sg2 and CSE2-sg3 poplars with mutations of *CSE1* and *CSE2*, respectively (Figure 2).

Consistent with previous reports, CSE1-sg2 and CSE2-sg3 poplars had up to 29.1% reduced lignin deposition (Figure 3) [15,21,22]. In our analysis of stem anatomy (Figure 4b), we found both CSE1-sg2 and CSE2-sg3 poplars had collapsed xylem vessel formation with reduced Wiesner staining; as this stain reacts with O-4-linked coniferyl and sinapyl aldehydes in lignified cells [35], this further confirmed a reduction in lignin content. In addition, a decrease in S-lignin content was revealed by Mäule staining (Figure 4c), which specifically stains S units red [36,37,38]. This result is consistent with the previous finding that CSE proteins function after the branch where G and S unit biosynthesis diverges from that of H units in the lignin pathway [15,22].

### 3.2. CSE1-sg2 and CSE2-sg3 Poplars Exhibit Normal Growth Performance Based on a Long-Term Field Test

*CSE* loss-of-function mutants of *Arabidopsis* and *M. truncatula* displayed severe dwarfing and altered development [15,21]. However, hpCSE lines (CSE-RNAi silencing) of hybrid poplar did not have drastically altered plant growth or development even though these lines had up to 25% reduced lignin deposition [22]. The mild phenotype in the hpCSE lines is likely due to residual expression of both *PtxaCSE* paralogues [22]. However, because RNAi silencing simultaneously downregulates both *CSE* genes, it was difficult for these researchers to investigate the individual roles of each of the two genes.

In the hybrid poplar used in this study (*Populus alba* × *P. glandulosa*, clone BH), both *PagCSE1* (indicating *PaCSE1* and *PgCSE1*, together) and *PagCSE2* genes were strongly and preferentially expressed in mature developing xylem (MDX) tissue, whereas much lower transcript levels were detected in shoot apical meristem with leaf primordia (SL), intermediate or mature stem-derived cambium (IC or MC), and leaves without veins (ML) [39]. Therefore, if one of the two *CSE* genes is unavailable, it is very likely that the other can function as a paralog for lignin biosynthesis. Indeed, our CSE1-sg2 and CSE2-sg3 poplars grew like control poplars, as demonstrated in our long-term LMO field test covering all four seasons (Figure 6b). This result can be explained by the fact that unlike in *Arabidopsis* and *M. truncatula*, only one of the two *CSE* genes was knocked out in the CSE1-sg2 and CSE2-sg3 poplars, respectively. Furthermore, *PagCSE1* and *PagCSE2* appear to be functional paralogs in our hybrid poplar.

### 3.3. CSE-Knockout Improves the Saccharification Efficiency of Poplar Stems

It has been well documented that lignin is a major impediment to the conversion of plant biomass into fermentable sugars [6,34]. To produce economically feasible biofuels, many efforts have been made to reduce the recalcitrance of biomass feedstock due to lignin [40,41,42,43]. Reducing *CSE* function has been proven to produce better biomass feedstock by reducing the recalcitrance of *Arabidopsis* and hybrid poplar to high saccharification [15,22]. Very recently, de Vries et al. (2021) [44] reported CRISPR-Cas9 editing of *CSE* in *Populus tremula* × *P. alba*, an approach very similar to that used in this study. However, in their study, CRISPR-Cas9-generated *cse1* and *cse2* single mutants had no significant phenotype and a wild-type lignin level; only *cse1 cse2* double mutants showed a reduction in lignin (35%) with a severe growth penalty. The *cse1 cse2* double mutants had a four-fold increase in cellulose-to-glucose conversion upon limited saccharification [44].

Unlike the report of de Vries et al. (2021) [44], our CSE1-sg2 and CSE2-sg3 poplars had significantly reduced lignin levels (up to 29.1%) and thus showed a dramatic increase in saccharification efficiency (Figure 6a). It is not clear why the results are different at this point, but perhaps the different species and the different target sites of CRISPR might also be the reason. Additionally, because the hpCSE line had no growth penalty with a 25% reduction in lignin [22], no phenotypic effect is likely as long as the amount of lignin remains above a certain threshold.

Although the saccharification efficiency of the CSE1-sg2 and CSE2-sg3 poplars was lower than that of *cse1 cse2* double mutant poplars [44], there was no associated growth penalty and, thus, CSE1-sg2 and CSE2-sg3 transgenic poplars can be directly utilized as efficient biomass feedstock for biorefineries.

## 4. Materials and Methods

### 4.1. Plant Materials and Growth Conditions

Hybrid poplars (*Populus alba* × *P. glandulosa*, clone BH) were used as both wild-type controls and transgenic plants in this study. Plants were acclimated in soil and grown in a growth room (16 h light; light intensity, 150 µmol m^−2^s^−1^; 24 °C) or in an LMO field at the Forest Bioresources Department of the National Institute of Forest Science, Republic of Korea (latitude 37.2 N, longitude 126.9 E).

### 4.2. Growth Measurements

Stem height was measured using a scale bar from the top of the plant to the soil level, and stem diameter was measured using digital calipers (Mitutoyo, Kawasaki, Japan) at 3 cm above soil level. Three biological replicates per line were analyzed.

### 4.3. CSE-CRISPR/Cas9 Vector Construction and Plant Transformation

Single guide RNAs (sgRNAs) targeting *CSE* genes were designed by Cas-Designer in the CRISPR RGEN Tools (http://www.rgenome.net/cas-designer/20210817) using full-length cDNA sequences of *CSE* genes (i.e., *PaCSE1*, *PgCSE1*, *PaCSE2* and *PgCSE2*) and the *Populus alba* × *P. tremula* var. glandulosa (Poplar 84K) genome as a reference sequence. Target sequences were selected with a low expected number of mismatches and high out-of-frame score (Appendix A). Finally, three single guide RNAs (sg1–sg3) were selected for knockout of *CSE1*, *CSE2*, or both genes, and each guide RNA length was set to 20 bp excluding the protospacer adjacent motif (PAM) sequence (Appendix A). The binary vector pHAtC (GenBank: KU213971.1) and *Aar*I-mediated sgRNA cloning system [45] were used for *Agrobacterium*-mediated transformation of the hybrid poplar. In brief, the annealed target sgRNA sequence was inserted between the AtU6 promoter and sgRNA scaffold after *Aar*I-digestion and then circularized by T4 DNA ligase (New England Biolabs, Ipswich). Vector construct was then introduced into *Agrobacterium tumefaciens* strain GV3101, which was used to transform poplar using the stem node transformation–regeneration method [46,47]. All constructs used in this study were verified by DNA sequencing (Macrogen http://dna.macrogen.com/kor/20210818).

### 4.4. Genotyping of Regenerated Transgenic Hybrid Poplars by Targeted Deep Sequencing

Genotyping of the mutated sequences in transgenic hybrid poplars was performed using the Illumina MiniSeq platform (KAIST Biocore Center, Daejeon, Korea). In brief, genomic DNA was extracted from shoot tissue of regenerated transgenic hybrid poplars using the DNeasy Plant Mini Kit (Qiagen, Hilden, Germany). The target region was amplified using nested PCR primer pairs containing adapter sequences. Then, amplicons were labelled with an index sequence (Illumina, Seoul, Korea) using index PCR primer pairs, and targeted deep sequencing was conducted using an Illumina MiniSeq (KAIST Biocore Center, Daejeon, Korea). The resulting deep sequencing data were analyzed using Cas-Analyzer (www.rgenome.net/cas-analyzer/20210817). Primer pairs used in this study are listed in Appendix A.

### 4.5. Histological Analysis

Cross sections of poplar stems were prepared by hand-cutting and stained with 0.05% toluidine blue *O* or 2% phloroglucinol/HCl for 1 min, as described previously [48]. Mäule staining was performed following the method of Mitra and Loqué [49]. In brief, stem cross sections were incubated for 2 min in 1 mL of 0.5% (*w*/*v*) potassium permanganate. Sections were then rinsed with distilled water 3–4 times until the solution remained clear. Then, 1 mL of 3% HCl was added to remove the deep brown color of the stained sections. The 3% HCl solution was removed and 1 mL of 14.8 M ammonium hydroxide solution was added immediately. Sections were observed using a digital camera-equipped microscope (CHB-213; Olympus, Tokyo, Japan).

### 4.6. RNA Extraction and RT-qPCR

For RNA extraction of hybrid poplars, the cetyltrimethylammonium bromide (CTAB) method was used because of the high amounts of polysaccharides and polyphenols in poplars, as described previously [48,50]. One microgram of total RNA was reverse transcribed using Superscript III reverse transcriptase (Invitrogen, Carlsbad, CA, USA) in a 20 μL reaction volume. Subsequently, RT-PCR was performed using 1 μL of the reaction product as a template. Quantitative real-time PCR was performed using an CFX96 Touch ™ Real-Time PCR platform (BioRad) with iQ^TM^ SYBR^®^ Green Supermix (BioRad, Hercules, CA, USA). Poplar *ACTIN7* (Potri.001G309500) was used as the internal quantitative control, and relative expression level was calculated by the 2^−ΔΔ^^CT^ method [51]. All primer sequences were designed using Primer3 software (http://fokker.wi.mit.edu/20210807). Sequences are provided in Appendix A.

### 4.7. Measurement of Klason Lignin Content

Klason lignin (i.e., acid insoluble lignin) contents of transgenic poplars grown for 3 months in soil were measured [52]. Stem tissues were dried at 65 °C for 1 week and ground to a fine powder. Ground materials (~100 mg) were placed in glass screw-cap tubes and 1 mL of 72% (*v*/*v*) sulfuric acid was added followed by thorough mixing. Tubes were placed in a water bath set at 45 ± 3 °C and incubated for 90 ± 5 min until all samples were hydrolyzed. Acid was diluted to a 4% concentration by adding 28 mL deionized water. Samples were mixed by inversion several times to eliminate phase separation. Sealed samples were autoclaved for 1 h at 121 °C and slowly cooled down to room temperature before removing the caps of the tubes. The autoclaved hydrolysis solution was vacuum-filtered through pre-weighed filter paper. The filter paper was dried at 105 °C to obtain acid insoluble residue until a constant weight was achieved. The filter paper was allowed to cool down to room temperature and the weight of the filter paper and dry residue were recorded.

### 4.8. Cell Wall Composition Analysis

The main stems of 8-month-old LMO field-grown hybrid poplars were used for cell wall composition analysis. Stem tissues were dried (65 °C/2 weeks) and ground to a fine powder. To determine extractives amounts [53], 50 mL of acetone was added to 700 mg of samples followed by a 2-hr incubation at 65 °C with shaking. After vacuum filtration and washing (5 mL of 10% (*v*/*v*) acetone three times), the filter paper was dried in an oven at 65 °C until a constant weight was obtained, which was then recorded. To extract hemicellulose [54], 4 mL of 10% (*w*/*v*) NaOH was added to 200 mg of the collected extractive-free samples above followed by a 3 h incubation at 50 °C with shaking. After vacuum filtration and washing (5 mL of distilled water three times), samples were dried (65 °C) until a constant weight was obtained, and the final weight of residue was recorded. Lignin content was determined using the Klason lignin method [52]. Cellulose content was obtained by calculating the difference between the initial samples (100%) and the percentages of the three other components.

### 4.9. Saccharification Efficiency of Transgenic Poplar

Saccharification efficiency was measured as described previously [50] with determination of reducing sugar content by the method of Yang et al. [55] with slight modifications. Briefly, for pretreatment, ground materials (~2 mg) were transferred into 2-mL screw-cap tubes and incubated with 200 μL of distilled water or 180 μL of NaOH (1%, *w*/*v*) at 30 °C for 30 min and then autoclaved at 120 °C for 60 min. After cooling to room temperature, 20 μL of 2.5 N HCl was used to neutralize the 1% NaOH-treated sample. After pretreatment, 300 μL of 0.1 M sodium acetate buffer (pH 5.0) containing 40 μg of tetracycline, 10 mg cellulose, and 1 mg ß-glucosidase was added. After 24, 48, and 72 h of incubation at 37 °C with shaking (180 rpm), samples were centrifuged (15,000× *g* for 3 min) and 5 μL of the supernatant was collected to measure reducing sugar content using the DNS (3,5-dinitrosalicylate) assay [56]. DNS reactions were performed by mixing 5 μL of the sample and 5 μL of water with 90 μL of DNS reagent in a PCR tube, followed by incubation at 95 °C for 6 min. Reducing sugar content was quantified by measuring the absorbance at λ_550 nm_ with glucose solution standards.

### 4.10. Statistical Analysis

All experiments were performed in triplicate and repeated at least three times. The number of used plants is indicated for each result presented. Statistical analyses were performed and graphs were generated using SigmaPlot v12.0 (Systat Software, Inc., Chicago, IL, USA). In addition, the significance of differences was calculated using Student’s *t*-test.

## Figures and Tables

**Figure 1 ijms-22-09750-f001:**
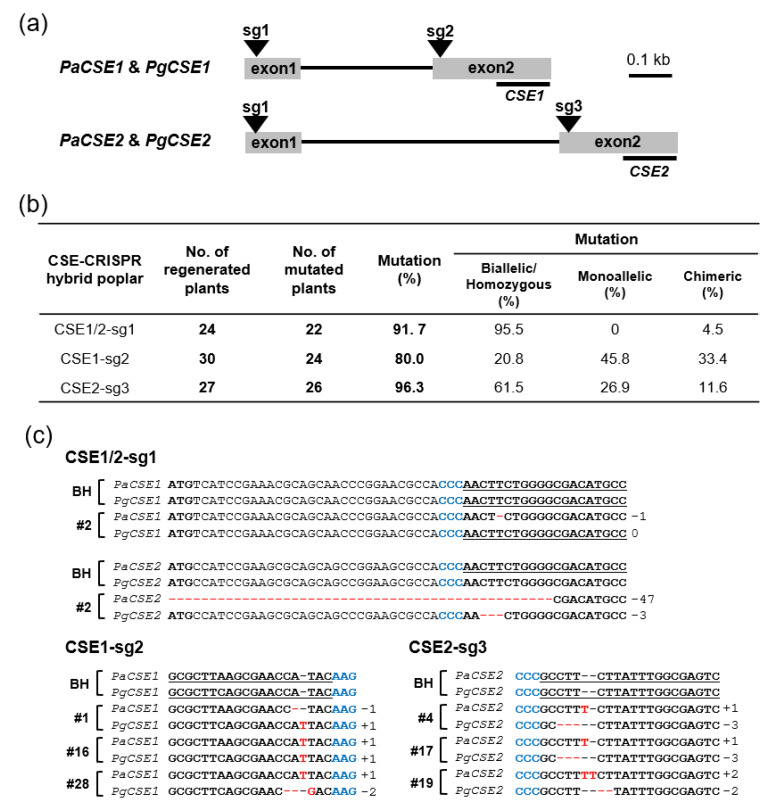
Production of transgenic hybrid poplar with CRISPR-knockout of *CSE* genes. (**a**) Gene structure of *CSE* genes with locations of single guide RNA (sg1–sg3) target sequences. *PaCSE1/2* and *PgCSE1/2* indicate *CSE* genes from *Populus alba* and *P. glandulosa*, respectively, in the hybrid poplar (*Populus alba* × *P. glandulosa*, clone BH) used in this study. C-terminal region of exon2 in each CSE gene for quantitative real-time PCR (RT-qPCR) target was underlined with a gene name. (**b**) Summary of the genotypes of the transgenic CSE-CRISPR hybrid poplars by targeted deep sequencing. Genotyping of the mutated sequences in transgenic hybrid poplars was performed using the Illumina MiniSeq platform (see, Methods). (**c**) Indel mutations of the selected CSE-CRISPR hybrid poplars. CSE1/2-sg1 (line #2) targeted the first exon of *PaCSE1/2* and *PgCSE1/2* genes; CSE1-sg2 (lines #1, #16, #28) targeted the second exon of *PaCSE1* and *PgCSE1*; CSE2-sg3 (lines #4, #17 #19) targeted the second exon of *PaCSE2* and *PgCSE2*. PAM sequences are shown in blue and target sequences are underlined. Identified indels in target sequences of each line are highlighted in red and the indel numbers are shown on the right.

**Figure 2 ijms-22-09750-f002:**
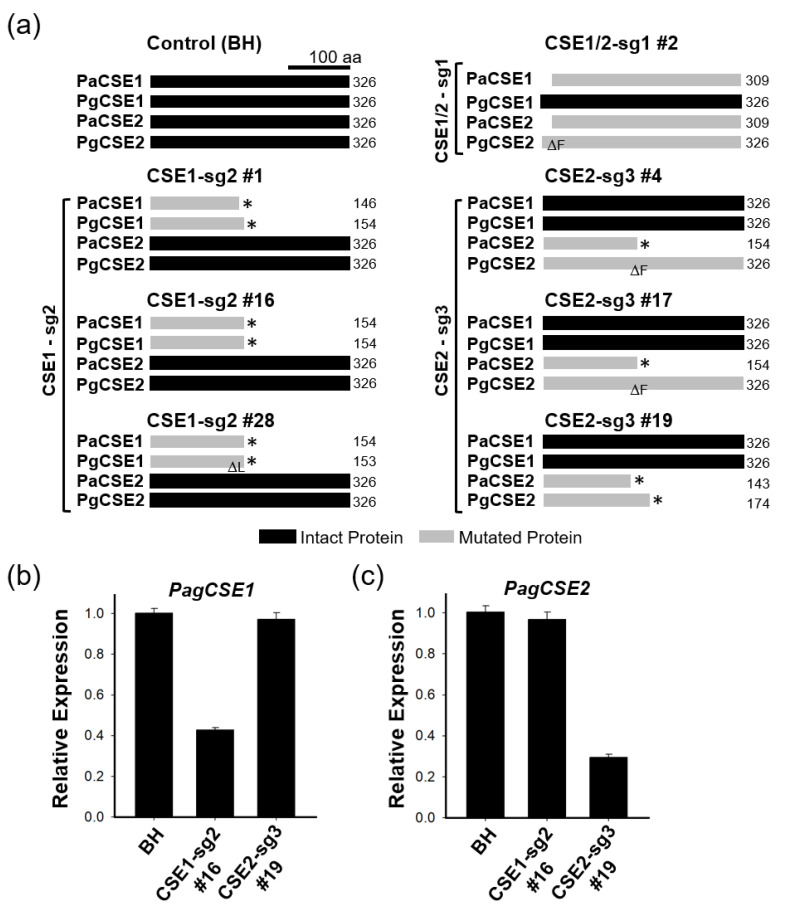
Predicted CSE protein and *CSE* gene expression in transgenic CSE-CRISPR hybrid poplars. (**a**) Summary of mutations in each transgenic CSE-CRISPR poplar line. Expected normal CSE proteins in the control (BH) are shown as black bars with corresponding sizes. CSE proteins mutated by CRISPR editing are shown as gray bars with corresponding sizes. Asterisks indicate non-sense mutations, and means a deletion of an amino acid. (**b**,**c**) Expression of *CSE* gene in CSE-CRISPR poplars. Quantitative real-time PCR (RT-qPCR) was performed using primers targeting the c-terminal regions shown in (Figure 1a) (*n* = 6, error bar = S.E.). *PagCSE1* indicates both *PaCSE1* and *PgCSE1*.

**Figure 3 ijms-22-09750-f003:**
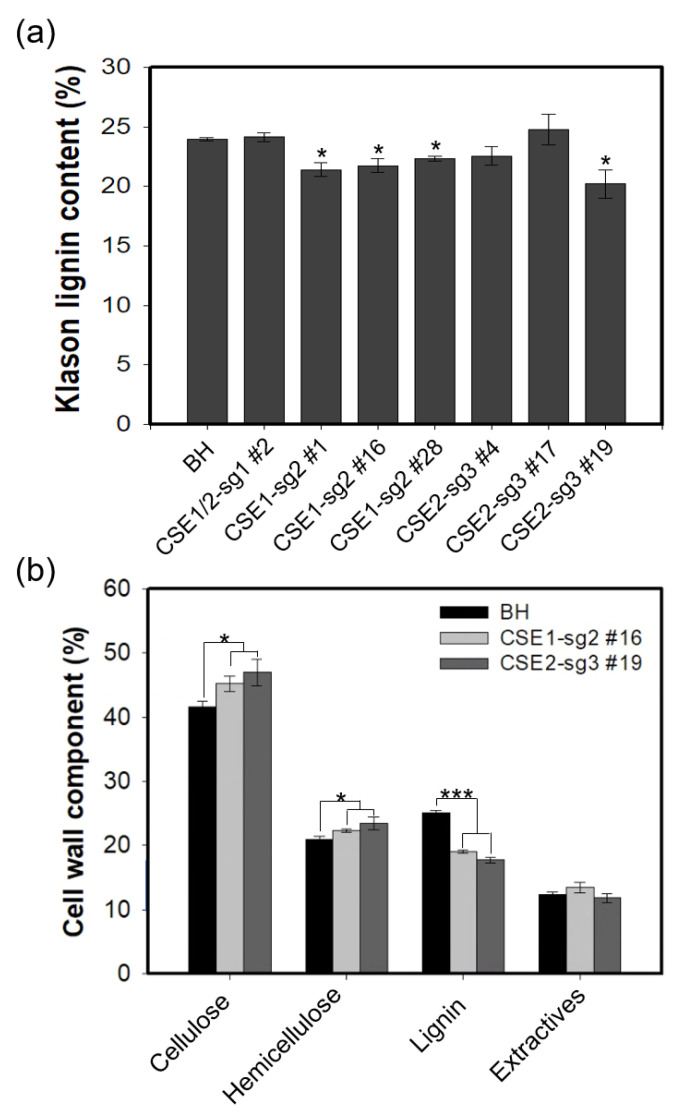
Transgenic CSE-CRISPR hybrid poplars have reduced lignin deposition. (**a**) Quantification of Klason lignin content. Two hundred milligrams of cell wall material from three-month-old hybrid poplars grown in pot was used (*n* = 2, error bar = S.E.). (**b**) Cell wall composition analysis. Eight-month-old LMO field grown stem tissues were used to analyze the composition of cell wall components (*n* = 3, error bar = S.E.). Asterisks indicate significant differences compared to BH using the unpaired Student’s *t*-test (* *p*-value < 0.05, *** *p*-value < 0.001).

**Figure 4 ijms-22-09750-f004:**
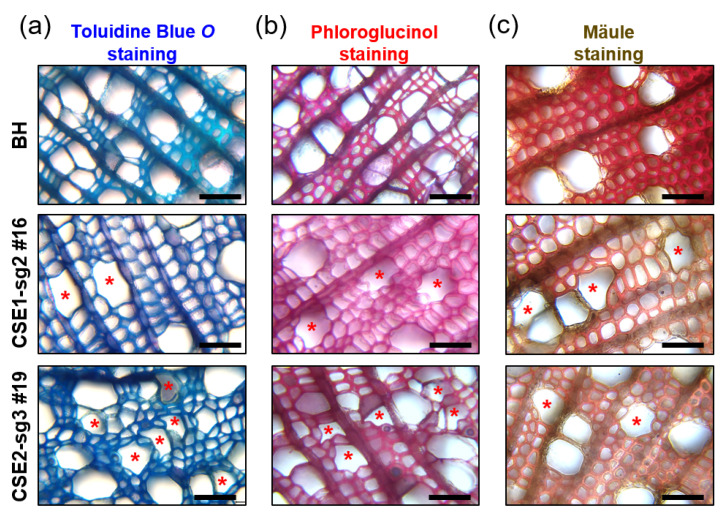
Transgenic CSE-CRISPR hybrid poplars have irregularly shaped xylem vessel cells. Stem anatomy of hybrid poplars (8-month-old LMO field grown) was assessed by (**a**) toluidine blue, (**b**) phloroglucinol-HCl, and (**c**) Mäule staining. Collapsed irregular vessels are marked with asterisks. Scale bars represent 50 µm.

**Figure 5 ijms-22-09750-f005:**
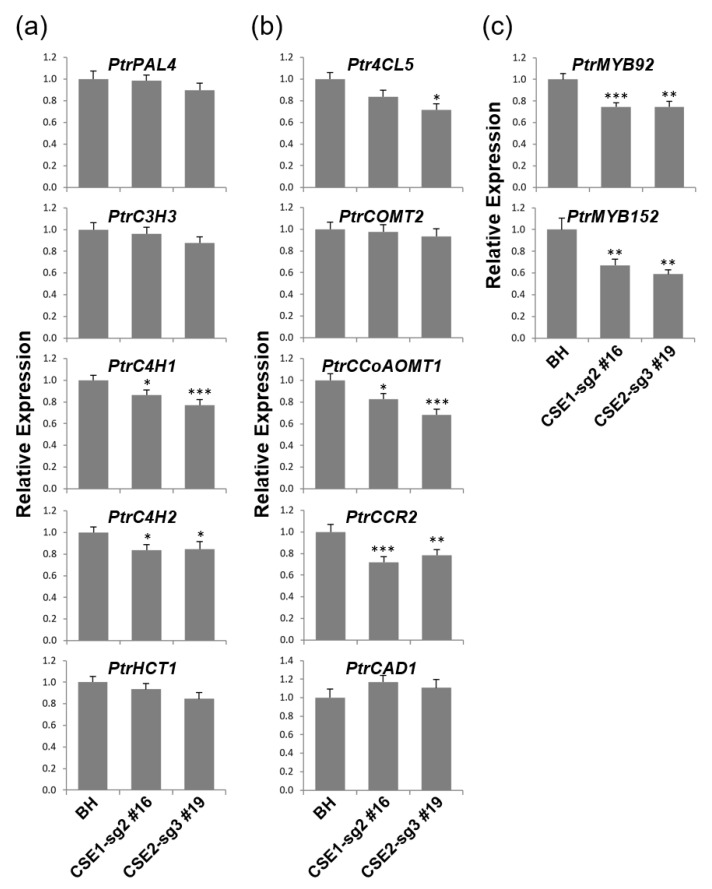
Expression of genes involved in lignin biosynthesis in transgenic CSE-CRISPR hybrid poplars. (**a**) Genes in the early steps of the lignin biosynthesis pathway: PtrPAL4 (Potri.010G224100.1); PtrC3H3 (Potri.006G033300.1); PtrC4H1 (Potri.013G157900.1); PtrC4H2 (Potri.019G130700.1); PtrHCT1 (Potri.003G183900.1). (**b**) Genes in the later steps of the lignin biosynthesis pathway: Ptr4CL5 (Potri.003G188500.1); PtrCOMT2 (Potri.012G006400.1); PtrCCoAOMT1 (Potri.009G099800.1); PtrCCR2 (Potri.003G181400.1); PtrCAD1 (Potri.009G095800.1). (**c**) Transcriptional regulation: PtrMYB92 (Potri.001G118800.1) and PtrMYB152 (Potri.017G130300.1). Relative expression (log_2_ scale) were determined by RT-qPCR using the *PtrACTIN7* gene as a quantitative control. Total RNA was extracted from the stems of 4-month-old poplars. Asterisks indicate significant differences compared to BH using the unpaired Student’s *t*-test (* *p*-value < 0.05, ** *p*-value < 0.01, *** *p*-value < 0.001). Error bars indicate the standard errors of three independent experiments.

**Figure 6 ijms-22-09750-f006:**
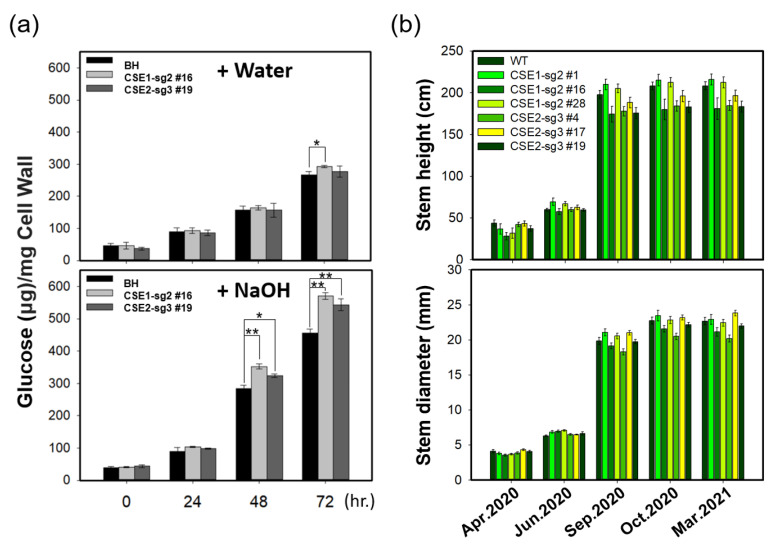
Enhanced saccharification efficiency of transgenic CSE-CRISPR hybrid poplars with normal growth. (**a**) Measurement of saccharification efficiency from BH and transgenic CSE-CRISPR hybrid poplars. Saccharification efficiency was estimated by analyzing the glucose content produced by cell wall materials of woody stems of poplar plants after water (upper panel) or NaOH (lower panel) treatment for the indicated times (see Methods). Error bars indicate S.E. (*n* = 8) (hour = hr.). Unpaired Student’s *t*-test, *p*-value * (*p* < 0.05), ** (*p* < 0.01), (**b**) Measurement of stem height and diameter at the indicated time (*n* = 20, Error bars = S.E.).

## Data Availability

All data are available from the corresponding author upon reasonable request.

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
