# Peer review of "CRISPR-Knockout of *CSE* Gene Improves Saccharification Efficiency by Reducing Lignin Content in Hybrid Poplar"

_ijms, 2021, doi:10.3390/ijms22189750_

Round 1
Reviewer 1 Report
The article is very interesting and informative. I have very few suggestions.
I suggest - CRISPR-knockout of CSE gene improves saccharification efficiency by reducing lignin content in hybrid poplar
“In both Arabidopsis and hybrid poplar, saccharification efficiency can be dramatically increased by mutation of CSE, suggesting that CSE is not only important for lignin biosynthesis but is also a promising target for generating improved lignocellulosic biomass crops for biofuel production [15, 22].” – this is very short and abruptly ending, better to elaborate on present understanding and genes known in the pathway.
Results
Authors have not shown evaluation of plants with the heterologous conditions at one, both, or in different combinations of knockout mutations in the homologs.
Author Response
The article is very interesting and informative. I have very few suggestions.
-- Authors’ response: Thank you very much for taking the time to evaluate this manuscript (IJMS-1367102) and your positive and excellent comments. We tried our best to improve our manuscript by reflecting your comments and hope this revision meets your expectations. Thanks again and have a great day!
Comment #1: I suggest - CRISPR-knockout of CSE gene improves saccharification efficiency by reducing lignin content in hybrid poplar
Response #1: Thanks for your suggestion. We love to change the title as you suggested “CRISPR-knockout of CSE gene improves saccharification efficiency by reducing lignin content in hybrid poplar”.
Comment #2: “In both Arabidopsis and hybrid poplar, saccharification efficiency can be dramatically increased by mutation of CSE, suggesting that CSE is not only important for lignin biosynthesis but is also a promising target for generating improved lignocellulosic biomass crops for biofuel production [15, 22].” – this is very short and abruptly ending, better to elaborate on present understanding and genes known in the pathway.
Response #2: Thanks for your suggestion. We elaborated this sentence in this revised manuscript as “In both Arabidopsis and hybrid poplar, saccharification efficiency can be dramatically increased by mutation of CSE, due to the reduction of lignin deposition. However, the overall plant growth was not severely inhibited [15, 22]. These results suggest that CSE is not only important for lignin biosynthesis but is also a promising target for generating improved lignocellulosic biomass crops for biofuel production [15, 22].” (Lines 87-91).
Comment #3: Results; Authors have not shown evaluation of plants with the heterologous conditions at one, both, or in different combinations of knockout mutations in the homologs.
Response #3: Thanks for your excellent suggestion. More detailed evaluations in heterologous conditions will be at the top of our list of further study.
Reviewer 2 Report
Molecular Sciences Manuscript #: ijms-1367102
Authors: Hyun-A. Jang, et al.
Title: CRISPR-Knockout of CAFFEOYL SHIKIMATE ESTERASE in a hybrid poplar improves saccharification efficiency by reducing lignin content
The authors have presented evidence on impact of CRISPR/Cas9 knockout mutations of caffeoyl shikimate esterase (cse) genes on lignin production, cell wall components, and stem growth in Poplar hybrid variety (Populus alba x Populus glandulosa). The authors present evidence of the specific CRISPR-induced mutations (indels) on the four different cse genes in several transgenic plant lines. They quantify cse mRNA accumulation (RT qPCR) as well as lignin and cell wall components in the mutants, and all of these data indicate that the CRISPR indel mutants have varying levels of decrease in cse mRNA and lignin as well as partial collapse of xylem vessel cells, consistent with reduced lignin in secondary cell walls. These also correlate with increases in small but significant increases in saccharification (glucose liberation) after NaOH-treated cell walls, suggesting these would be a better source of biomass for biofuels and biomaterials.
In general, the data provided by the authors supports their conclusions. There are some relatively minor issues about the presented data that need to be addressed. Furthermore, there are some minor writing issues that also need to be resolved.
General scientific concerns about writing, data and conclusions.
One key scientific issue deals with what seems like is inconsistency between the results and data for the CSE1/2-sg1 #2 plant, in which three of the four cse genes have mutations yet there is not a measureable phenotype in this plant. In contrast, the CSE1-sg2 (#1, #16, and #28) and CSE2-sg3 (#19) are mutated in only two of the four cse genes, yet these mutants with what fewer mutated genes have a significant decrease in the cse mRNAs (Fig. 2). Again, this seems somewhat unexpected that fewer mutated genes and with what seems to less severe indel mutations have large mutant phenotypes. The authors do address this issue to some extent in the Results (lines 120-123) and bring up what seems like might be legitimate points and again in the Discussion (lines 212-217). Their argument about the N-terminus targeting of the CSE protein in CSE1/2-sg1 would be strengthened if the authors could cite work/publications on what might be known about biochemistry of CSE’s and if the N-terminus is important for this.
Related to the above question about the one CSE1/2-sg1 plant (#2) mutant plant and why did not have an observable phenotype, according to Figure 1b, there were a total of 22 mutant regenerated plants for this CRISPR knockout, yet only one plant (#2) is mentioned in the paper. Where any of the other CSE1/2-sg1 plants sequenced and tested for phenotypes? If not, why not? Please explain in Results.
A second general concern / question is about the cse mRNAs from other paralogous gene in the different mutants, related to Figure 2b. First, it would be good to include PagCSE1 and PagCSE2 mRNA levels from the CSE1/2-sg1 (#2) plant in Fig 2b. If lignin values are not decreased (as shown later in Figure 3), then knowing these mRNA levels, that would be predicted to be same as WT, would be a good test of that hypothesis. These are not shown. Please show, or explain why they are not.
Related, it would help to show the PagCSE2 mRNA in CSE1-sg2 mutant as well as PagCSE1 mRNA in CSE2-sg3 mutant. This would be important for it is possible there is a feedback regulation that might result in an increase in mRNA in these mutants. Knowing the levels of the paralogous mRNAs would be helpful. Please show these, or explain why they are not.
A third general group of issues are regarding the presented data and some inconsistencies in the statistical analysis and data presented across the different figures. Most of these issues are relatively minor and can be resolved through adjustments to the text. There are times it seems what is mentioned in text at times seems to only be the “most favorable” data and not necessarily full statistical data. See specific items in list below for these.
The fourth general group of issues that need to be resolved are around wording and questions about data and interpretations. Most of these seem relatively minor and should be resolvable with minor editing to the text. Again, see specific comments in list below for these.
Fifth general issue is regarding Supplemental Figure S3. It should be deleted for it presents data that have already be published by this group (Reference #39). Furthermore, Fig. S3 had data that lacked any statistics (Standard errors, etc…) such that to include it would require, as I see it, more data to provide the variance, number of samples, etc… Again, the best is to delete this supplemental figure that shows already published data. Be sure to also delete the citations to Fig. S3 in the text (Line 243, Line 387, and from supplemental files).
Final general issue. In the Discussion, it would seem important to discuss how these mutant plants might be used and for how long would they be grown for either biomaterial or biofuel production. Related to this, might is the prognosis of these lignin reduced (and partially collapsed xylem vessels cells) in a two – five year old tree? If the trees are collected for biomass after only a year, then it might not be an issue. But, if the trees would need to grow for two, three or more years, might there eventually be apparent phenotypes in taller trees with more mass that would rely on lignin to keep upright? These would be speculative, but might be important on a practical level.
List of specific issues that need to be corrected or addressed:
Abstract, Line 16: I would change the word “essential” to “important”. Essential implies that without it ( in a knocked out/deleted strain) all lignin production would stop. The data do not support this level. Furthermore, there is another cse-independent pathways for lignin production that forms the H Unit –based lignin, although this might be to a lesser extent.
Abstract, Line 19: A question for clarification. Is this hybrid (P. alba x P. glandulosa) the same as the hybrid that I have seen published elsewhere named (P. alba x P. tremula var. gladulosa)? If these are the same, be sure to use the most common published name.
Introduction, Line 47: The reference numbering is out of order. It should be, [6, 8, 10, 11].
Introduction, Line 87-88: refers to the “deletion of either CSE1 or CSE2 … This implies the full genes are deleted, which is not the case. Formally, these are a mix of indels. Best to be consistent and use that term throughout.
Results, Line 107: delete the “,” between “see” and “Methods)”.
Results, Lines 114-115: Authors mention that seven of the representative lines were further tested, but there is no explanation as to how these seven were selected from the 72 mutated plants (Fig. 1b). Furthermore, why was only one mutant plant of CSE1/2-sg1 further tested when three plants from both CSE1-sg2 and CSE2-sg3 were tested? This seems important in light of the fact that the one (#2) plant of CSE1/2-sg1 turned out to not have an observable phenotype.
Results, Figure 1c: First, in Fig. 1c under CSE1-sg2 for sequence of PgCSE1 in plant #28. The third dash in that sequence should be in “black” font (not red), since it is the same as BH.
Results, Figure 1c Legend: Please explain why the PAM sequence (shown in blue) is upstream (5’) of target sequence for CSE1/2-sg1 and CSE2-sg3 while in CSE1-sg2 it is downstream (3’) of the shown target sequence. This also goes for supplemental Figure S2b.
Results, Lines 134-136: For the RT-qPCR, the authors need to mention that Actin mRNA was used as an internal control for determining relative RNA expression. Also, this is not mentioned in the Methods section, which it should be, but Figure 5 Legend and the Table S1 mentions that actin and actin primers were used for control in RT-qPCR.
Results, Fig. 2b Legend: The “n” number of replicates and what the error bars show need to be included in this legend, similar to what was done for Figure 3 Legend.
Results, Lines 147 and 157 and Figures 3a vs 3b: In the text and in this figure, there are two different estimates of amount of reduction of lignin for the CSE1-sg2 and CSE2-sg3 mutants. In Figure 3a (and line 147), there is a claim of 16% decrease in lignin, presumably the authors are citing the CSE2-sg3 (#19) data for it is the most extreme while in Figure 3b (and line 157) there is a claim of a 29.1% decrease in lignin in the same mutant plants. First, why is there such a dramatic difference? Related, are the same replicate data presented in Figure 3a the same as the data presented in Figure 3b for lignin? It would appear they are not the same data, for in Figure 3a there was an “n = 2” while in Figure 3b the number of replicates (n) was 3. The biggest issue that the authors need to address is why the discrepancy in estimated decrease in lignin. Further, the authors cite only the highest value (29.1% decrease) in the Abstract (Line 24), yet the full data shows that the decrease ranges from 16 – 29.1%.
Results, Line 157-159: The indicate there is a small but significant increase in cellulose and hemicellulose. Since these are % values of cell wall components, if one component decreases, the others must (mathematically) increase even if there is not an absolute change in amount. It is still worth pointing this out in the Results, but be sure to mention these were increased on a percent of cell wall components basis.
Results, Line 178: Authors cite Fig. 5b and that Ptr4CL5 mRNA was “significantly suppressed in CSE-CRISPR poplars…” This is not completely accurate, for it was only significantly reduced in CSE2-sg3 (#19) and in CSE1-sg2 it was not significantly less than WT (according to Figure 5b data).
Results, Line 178: Authors cite Fig. 5b and that Ptr4CL5 mRNA was “significantly suppressed in CSE-CRISPR poplars…” This is not completely accurate, for it was only significantly reduced in CSE2-sg3 (#19) and in CSE1-sg2 it was not significantly less than WT (according to Figure 5b data).
Related, the text in this section (Results, Section 2.5) do not mention that two mRNAs in Figure 5a, for “early steps” in lignin synthesis also had significant decrease in expression in CRISPR mutants. These were PtrC4H1 and PtrC4H2 mRNAs. This also seems to go against what might be expected, if there were a feedback induction when lignin was lower. Thus, these seem worth pointing out in the Results and/or Discussion.
Results, Figure 6 Legend: The statistics shown here are inconsistent with other figure legends. In Figure 6 the authors show Standard Deviation (SD) for error bars while in all the rest of the figures they show Standard Error (SE), which always present as being smaller (based on how they are calculated). Frankly, SD would seem to be the more accurate statistics to show for all the figures shown in this manuscript. The reason it is particularly inappropriate to switch to SD for Figure 6, however, is because these data are those that one would hope to not see a difference in growth compared to WT. To avoid looking as if the statistics being shown are not based on what looks “most favorable”, be consistent in what is used (SD or SE) for error bars.
Results, Line 190. The authors cite a “25% increase in glucose from NaOH-treated” poplars. Is this from one of the specific CRISPR mutants or is this an average across the two different CRISPR mutants? More clarity on what these data represent is needed.
Discussion, Line 243: As already mentioned at “top”, delete Figure S3 for reasons mentioned above.
Supplemental Table S1: There is a formatting problem for the file I received. The words “Forward” and “Reverse” for the primers are presented in an awkward way. My guess is this is just a minor format result from uploaded figures for the review process. But, it should be confirmed that is the case.
References, Line 530-532: Delete Reference #57. It is a duplicate of Reference #39.
Author Response
The authors have presented evidence on impact of CRISPR/Cas9 knockout mutations of caffeoyl shikimate esterase (cse) genes on lignin production, cell wall components, and stem growth in Poplar hybrid variety (Populus alba x Populus glandulosa). The authors present evidence of the specific CRISPR-induced mutations (indels) on the four different cse genes in several transgenic plant lines. They quantify cse mRNA accumulation (RT qPCR) as well as lignin and cell wall components in the mutants, and all of these data indicate that the CRISPR indel mutants have varying levels of decrease in cse mRNA and lignin as well as partial collapse of xylem vessel cells, consistent with reduced lignin in secondary cell walls. These also correlate with increases in small but significant increases in saccharification (glucose liberation) after NaOH-treated cell walls, suggesting these would be a better source of biomass for biofuels and biomaterials. In general, the data provided by the authors supports their conclusions. There are some relatively minor issues about the presented data that need to be addressed. Furthermore, there are some minor writing issues that also need to be resolved.
-- Authors’ response: Thank you very much for taking time to evaluate our manuscript (IJMS-1367102) and your positive comments and excellent suggestions. We tried our best to improve our manuscript by reflecting your comments in this revised manuscript. Thanks to you, we believe the quality of our manuscript has been improved a lot. We hope this revision meets your expectations. Thanks again and have a great day!
General scientific concerns about writing, data and conclusions.
Comment #1: One key scientific issue deals with what seems like is inconsistency between the results and data for the CSE1/2-sg1 #2 plant, in which three of the four cse genes have mutations yet there is not a measureable phenotype in this plant. In contrast, the CSE1-sg2 (#1, #16, and #28) and CSE2-sg3 (#19) are mutated in only two of the four cse genes, yet these mutants with what fewer mutated genes have a significant decrease in the cse mRNAs (Fig. 2). Again, this seems somewhat unexpected that fewer mutated genes and with what seems to less severe indel mutations have large mutant phenotypes. The authors do address this issue to some extent in the Results (lines 120-123) and bring up what seems like might be legitimate points and again in the Discussion (lines 212-217). Their argument about the N-terminus targeting of the CSE protein in CSE1/2-sg1 would be strengthened if the authors could cite work/publications on what might be known about biochemistry of CSE’s and if the N-terminus is important for this.
Response #1: Thanks for your insightful discussion. As we discussed, although the CSE1/2-sg1 poplars have mutations in three genes out of four CSE genes, the CSE1/2-sg1 poplars are expected to have PagCSE1/2 proteins potentially function properly. The results of lignin content measurement (Figure 3a) support this notion. Unfortunately, however, there are no publications on CSE biochemistry yet to the best of our knowledge.
Comment #2: Related to the above question about the one CSE1/2-sg1 plant (#2) mutant plant and why did not have an observable phenotype, according to Figure 1b, there were a total of 22 mutant regenerated plants for this CRISPR knockout, yet only one plant (#2) is mentioned in the paper. Where any of the other CSE1/2-sg1 plants sequenced and tested for phenotypes? If not, why not? Please explain in Results.
Response #2: Thanks for pointing this out. Among 22 mutants generated, we performed in-depth analyses on a total of five lines (two biallelic (#11, #12) and three homo lines (#30, #32, #34). However, all those lines showed similar growth performance with no significant changes of lignin deposition (data not shown). To explain this, we added the following description in this revised manuscript as “In fact, we performed in-depth analyses on five additional lines, that are two biallelic (#11, #12) and three homo lines (#30, #32, #34). However, all those lines showed similar growth performances with no significant changes of lignin deposition compared to BH poplars (data not shown).” (Lines 242-245).
Comment #3: A second general concern / question is about the cse mRNAs from other paralogous gene in the different mutants, related to Figure 2b. First, it would be good to include PagCSE1 and PagCSE2 mRNA levels from the CSE1/2-sg1 (#2) plant in Fig 2b. If lignin values are not decreased (as shown later in Figure 3), then knowing these mRNA levels, that would be predicted to be same as WT, would be a good test of that hypothesis. These are not shown. Please show, or explain why they are not. Related, it would help to show the PagCSE2 mRNA in CSE1-sg2 mutant as well as PagCSE1 mRNA in CSE2-sg3 mutant. This would be important for it is possible there is a feedback regulation that might result in an increase in mRNA in these mutants. Knowing the levels of the paralogous mRNAs would be helpful. Please show these, or explain why they are not.
Response #3: Thanks for your excellent comments. We performed RT-qPCR again to check the PagCSE1 and PagCSE2 expression in CSE1-sg2 and CSE2-sg3 mutants with BH control poplar. Unfortunately, however, we could not get the PagCSE1 and PagCSE2 expression from CSE1/2-sg1 (#2) poplar because the samples are not available currently. The new result is provided in this revised manuscript as Figure 2b and c (see below). As we expected, the expression of PagCSE1 has not been changed significantly in the CSE2-sg3 mutant. The opposite is the same. We added following description in this revised manuscript as “However, as expected, there is no significant change in PagCSE1 expression in CSE2-sg3 poplar and vice versa (Figure 2b, c).” (Lines 154-155).
Comment #4: A third general group of issues are regarding the presented data and some inconsistencies in the statistical analysis and data presented across the different figures. Most of these issues are relatively minor and can be resolved through adjustments to the text. There are times it seems what is mentioned in text at times seems to only be the “most favorable” data and not necessarily full statistical data. See specific items in list below for these.
Response #4: We corrected the issue of inconsistencies in the statistical analysis in this revised manuscript by using SE instead of SD (e.g., SD of Fig.6 was changed to SE).
Comment #5: The fourth general group of issues that need to be resolved are around wording and questions about data and interpretations. Most of these seem relatively minor and should be resolvable with minor editing to the text. Again, see specific comments in list below for these.
Response #5: We answered for the specific comments in list below.
Comment #6: Fifth general issue is regarding Supplemental Figure S3. It should be deleted for it presents data that have already be published by this group (Reference #39). Furthermore, Fig. S3 had data that lacked any statistics (Standard errors, etc…) such that to include it would require, as I see it, more data to provide the variance, number of samples, etc… Again, the best is to delete this supplemental figure that shows already published data. Be sure to also delete the citations to Fig. S3 in the text (Line 243, Line 387, and from supplemental files).
Response #6: We removed the Figure S3 with all the related descriptions in this revised manuscript including the reference and the cited reference in the figure legend.
Comment #7: Final general issue. In the Discussion, it would seem important to discuss how these mutant plants might be used and for how long would they be grown for either biomaterial or biofuel production. Related to this, might is the prognosis of these lignin reduced (and partially collapsed xylem vessels cells) in a two – five year old tree? If the trees are collected for biomass after only a year, then it might not be an issue. But, if the trees would need to grow for two, three or more years, might there eventually be apparent phenotypes in taller trees with more mass that would rely on lignin to keep upright? These would be speculative, but might be important on a practical level.
Response #7: Thanks for your excellent discussion. In fact, we are very interested in the issue you raised. Therefore, we would like to continue to monitor the growth for years, and will report relevant findings when available.
List of specific issues that need to be corrected or addressed:
Specific issue #1: Abstract, Line 16: I would change the word “essential” to “important”. Essential implies that without it ( in a knocked out/deleted strain) all lignin production would stop. The data do not support this level. Furthermore, there is another cse-independent pathways for lignin production that forms the H Unit –based lignin, although this might be to a lesser extent.
Response #1: Thanks for your comment. We changed the ‘essential’ to ‘important’ in this revised manuscript (Line 28).
Specific issue #2: Abstract, Line 19: A question for clarification. Is this hybrid (P. alba x P. glandulosa) the same as the hybrid that I have seen published elsewhere named (P. alba x P. tremula var. gladulosa)? If these are the same, be sure to use the most common published name.
Response #2: Yes, We think the hybrid (P. alba x P. glandulosa) is the same to the hybrid (P. alba x P. tremula var. gladulosa). However, with our best knowledge, this hybrid (e.g., clone 84K) was derived from our country (South Korea) and we use the P. alba x P. glandulosa as an official name. For your information, a genome analysis of this hybrid was published, recently (An improved draft genome sequence of hybrid Populus alba × Populus glandulosa. Journal of Forestry Research, 2021, 32:1663–1672).
Specific issue #3: Introduction, Line 47: The reference numbering is out of order. It should be, [6, 8, 10, 11].
Response #3: Thanks. We corrected the order in this revised manuscript (Line 61).
Specific issue #4: Introduction, Line 87-88: refers to the “deletion of either CSE1 or CSE2 … This implies the full genes are deleted, which is not the case. Formally, these are a mix of indels. Best to be consistent and use that term throughout.
Response #4: Thanks. We changed the ‘deletion’ to ‘mutation’ in this revised manuscript (Line 100).
Specific issue #5: Results, Line 107: delete the “,” between “see” and “Methods)”.
Response #5: Thanks. We corrected this in this revised manuscript (Line 121).
Specific issue #6: Results, Lines 114-115: Authors mention that seven of the representative lines were further tested, but there is no explanation as to how these seven were selected from the 72 mutated plants (Fig. 1b). Furthermore, why was only one mutant plant of CSE1/2-sg1 further tested when three plants from both CSE1-sg2 and CSE2-sg3 were tested? This seems important in light of the fact that the one (#2) plant of CSE1/2-sg1 turned out to not have an observable phenotype.
Response #6: As we discussed above (response #2 to comment #2), we analyzed a total of 6 lines of CSE1/2-sg1 poplars. However, all lines have similar phenotype (Lines 236-239 in this revised manuscript). Thus, we are making new CRISPR lines targeting region in the middle of both CSE1 and CSE2 genes. And regarding the seven representative lines, we selected lines of higher indel biallelic/homozygous mutations as described in the manuscript (Lines 127-128).
Specific issue #7: Results, Figure 1c: First, in Fig. 1c under CSE1-sg2 for sequence of PgCSE1 in plant #28. The third dash in that sequence should be in “black” font (not red), since it is the same as BH.
Response #7: Thanks a lot. We corrected this in this revised manuscript.
Specific issue #8: Results, Figure 1c Legend: Please explain why the PAM sequence (shown in blue) is upstream (5’) of target sequence for CSE1/2-sg1 and CSE2-sg3 while in CSE1-sg2 it is downstream (3’) of the shown target sequence. This also goes for supplemental Figure S2b.
Response #8: Target sequences were designed by Cas-Designer of the CRISPR RGEN Tools (http://www.rgenome.net/cas-designer) as described in Methods and selected with a low expected number of mismatches and high out-of-frame score (Lines 311-316 in this manuscript). As you know, targeting either strand of DNA is fine in CRISPR/Cas9 technology.
Specific issue #9: Results, Lines 134-136: For the RT-qPCR, the authors need to mention that Actin mRNA was used as an internal control for determining relative RNA expression. Also, this is not mentioned in the Methods section, which it should be, but Figure 5 Legend and the Table S1 mentions that actin and actin primers were used for control in RT-qPCR.
Response #9: Thanks for pointing this. As you suggested, we added the description in this revised manuscript as “As an internal quantitative control, PtrACTIN7 (Potri. 001G309500) gene was used.” (Lines 151-152). However, we included this description in the Method section in our original manuscript (Lines 365-366; Poplar ACTIN7 (Potri.001G309500) was used as the internal quantitative control, and relative expression level was calculated by the 2-ΔΔCT method [51]).
Specific issue #10: Results, Fig. 2b Legend: The “n” number of replicates and what the error bars show need to be included in this legend, similar to what was done for Figure 3 Legend.
Response #10: Thanks again. We added ‘(n = 6, error bar = S.E.)’ in this revised manuscript.
Specific issue #11: Results, Lines 147 and 157 and Figures 3a vs 3b: In the text and in this figure, there are two different estimates of amount of reduction of lignin for the CSE1-sg2 and CSE2-sg3 mutants. In Figure 3a (and line 147), there is a claim of 16% decrease in lignin, presumably the authors are citing the CSE2-sg3 (#19) data for it is the most extreme while in Figure 3b (and line 157) there is a claim of a 29.1% decrease in lignin in the same mutant plants. First, why is there such a dramatic difference? Related, are the same replicate data presented in Figure 3a the same as the data presented in Figure 3b for lignin? It would appear they are not the same data, for in Figure 3a there was an “n = 2” while in Figure 3b the number of replicates (n) was 3. The biggest issue that the authors need to address is why the discrepancy in estimated decrease in lignin. Further, the authors cite only the highest value (29.1% decrease) in the Abstract (Line 24), yet the full data shows that the decrease ranges from 16 – 29.1%.
Response #11: Thanks for raising this issue. We think the differences come from the different growth condition and age of hybrid poplars. Stem samples of Figure 3a were THREE-month-old poplars grown in pot under the greenhouse condition while the stem samples of Figure 3b were EIGHT-month-old poplars grown in LMO field as described in the figure legend. We believe the latter case is closer to the real value. And the Figure 3a is more like a preliminary screening purpose. To clarify this, we added the following description in this revised manuscript “This reduction in lignin content is higher than the results in Figure 3a, which may result from different growth conditions (e.g., 3 months in pots vs. 8 months in LMO fields).” (Lines 175-177).
Specific issue #12: Results, Line 157-159: The indicate there is a small but significant increase in cellulose and hemicellulose. Since these are % values of cell wall components, if one component decreases, the others must (mathematically) increase even if there is not an absolute change in amount. It is still worth pointing this out in the Results, but be sure to mention these were increased on a percent of cell wall components basis.
Response #12: Thanks for this comment. We will keep that in mind.
Specific issue #13: Results, Line 178: Authors cite Fig. 5b and that Ptr4CL5 mRNA was “significantly suppressed in CSE-CRISPR poplars…” This is not completely accurate, for it was only significantly reduced in CSE2-sg3 (#19) and in CSE1-sg2 it was not significantly less than WT (according to Figure 5b data).
Response #13: To avoid misunderstanding, we removed the ‘Ptr4CL5’ in Line 200 in this revised manuscript.
Specific issue #14: Related, the text in this section (Results, Section 2.5) do not mention that two mRNAs in Figure 5a, for “early steps” in lignin synthesis also had significant decrease in expression in CRISPR mutants. These were PtrC4H1 and PtrC4H2 mRNAs. This also seems to go against what might be expected, if there were a feedback induction when lignin was lower. Thus, these seem worth pointing out in the Results and/or Discussion.
Response #14: Thanks for this detail. Again, to avoid misunderstanding, we modified the description in this revised manuscript as “As expected, genes upstream of CSE showed relatively stable expression levels compared to downstream genes, except PtrC4H1 and PtrC4H2 genes (Figures 5a, b).” (Lines 198-199).
Specific issue #15: Results, Figure 6 Legend: The statistics shown here are inconsistent with other figure legends. In Figure 6 the authors show Standard Deviation (SD) for error bars while in all the rest of the figures they show Standard Error (SE), which always present as being smaller (based on how they are calculated). Frankly, SD would seem to be the more accurate statistics to show for all the figures shown in this manuscript. The reason it is particularly inappropriate to switch to SD for Figure 6, however, is because these data are those that one would hope to not see a difference in growth compared to WT. To avoid looking as if the statistics being shown are not based on what looks “most favorable”, be consistent in what is used (SD or SE) for error bars.
Response #15: We corrected the issue of inconsistencies in the statistical analysis in this revised manuscript. For example, the SD of Fig.6 was changed to SE.
Specific issue #16: Results, Line 190. The authors cite a “25% increase in glucose from NaOH-treated” poplars. Is this from one of the specific CRISPR mutants or is this an average across the two different CRISPR mutants? More clarity on what these data represent is needed.
Response #16: We modified the description in this revised manuscript as “We found a significant increase (>25% at 72 h) in glucose release from NaOH-treated CSE-CRISPR poplars (CSE1-sg2 #16) compared to BH poplars (Figure 6a).” (Lines 213-215).
Specific issue #17: Discussion, Line 243: As already mentioned at “top”, delete Figure S3 for reasons mentioned above.
Response #17: We removed it in this revised manuscript as mentioned above (Response #6 to Comment #6).
Specific issue #18: Supplemental Table S1: There is a formatting problem for the file I received. The words “Forward” and “Reverse” for the primers are presented in an awkward way. My guess is this is just a minor format result from uploaded figures for the review process. But, it should be confirmed that is the case.
Response #18: We reformatted the Supplemental Table S1 in this revised manuscript.
Specific issue #19: References, Line 530-532: Delete Reference #57. It is a duplicate of Reference #39.
Response #19: Thanks. We removed this reference in this revised manuscript.
